# Leucine-Rich Alpha-2-Glycoprotein: A Novel Predictor of Diastolic Dysfunction

**DOI:** 10.3390/biomedicines11030944

**Published:** 2023-03-20

**Authors:** Alexander Loch, Kok Leng Tan, Mahmoud Danaee, Iskandar Idris, Mei Li Ng

**Affiliations:** 1Department of Medicine, Faculty of Medicine, University of Malaya, Kuala Lumpur 50603, Malaysia; alexanderloch@gmx.de (A.L.); mdanaee@um.edu.my (M.D.); 2Advanced Medical and Dental Institute, Universiti Sains Malaysia, Kepala Batas 13200, Malaysia; tankl@usm.my; 3Division of Medical Sciences and Graduate Entry Medicine, School of Medicine, University of Nottingham, Nottingham NG7 2UH, UK; iskandar.idris@nottingham.ac.uk; 4Department of Medicine, National University of Singapore, Singapore 119228, Singapore

**Keywords:** leucine-rich alpha-2-glycoprotein 1 (LRG1), diastolic dysfunction, biomarker, heart failure with preserved ejection fraction (HFpEF), cross-sectional study

## Abstract

Leucine-rich α2-glycoprotein (LRG1) mediates cardiac fibrocyte activation. It is upregulated in inflammatory conditions, atherosclerosis, and fibrosis. Diastolic dysfunction (DD) is due to myocardial fibrosis. This cross-sectional study examined the relationship between LRG1 and DD. Patients with symptoms of chronic coronary ischemia were recruited. Patients with symptoms of overt heart failure, ejection fraction (EF) < 55%, impaired renal function, infection, and recent trauma were excluded from the study. Clinical parameters examined were SYNergy between percutaneous coronary intervention with TAXus and cardiac surgery (SYNTAX) score, echocardiographic assessment, and LRG1 levels. Binary stepwise logistic regression was used to evaluate the association between LRG1 and DD. Receiver Operating Characteristic (ROC) analysis was used to determine optimal cut-off values and predictive performance of LRG1. A total of 94 patients were enrolled in the study, with 47 having a clinical diagnosis of DD. Plasma LRG1 was significantly (U = 417.00, *p* < 0.001) higher in the DD group (M = 14) compared to the No-DD group (M = 8) by Mann–Whitney U test. There were higher SYNTAX scores in the DD group (M = 24.5) compared with No-DD (M = 7). LRG1 had significant predictability of DD (OR = 1.32 (95% CI: 1.14–1.53)). The ROC showed an AUC = 0.89 (95% CI: 0.82–0.95). LRG1 had a 78% sensitivity (95% CI: 65.3–87.7) and 72.3% specificity (95% CI: 57.4–84.4) for predicting DD at a cut-off value of “9”. In conclusion, we identified LRG1 as a novel independent predictor of DD. Further studies are warranted to validate the utility of LRG1 in predicting DD.

## 1. Introduction

Heart failure with preserved ejection fraction (HFpEF) accounts for about half of all heart failure (HF) diagnoses, with morbidity and mortality that is on par with HF with reduced EF (HFrEF) [1]. The presence of diastolic dysfunction (DD) is a condition sine qua non to diagnose HFpEF. Diastolic dysfunction refers to the inability of the ventricle to accommodate blood from the atrium due to increased stiffness and reduced compliance. Asymptomatic diastolic dysfunction commonly progresses to symptomatic HFpEF [2].

Multiple mechanisms have been proposed with regard to the pathogenesis of diastolic dysfunction, including inflammation with increased interstitial deposition of collagen and matricellular proteins and chronic myocardial ischemia [3,4,5]. Patients diagnosed with coronary artery disease (CAD) have been shown to have a significantly higher risk for diastolic dysfunction [6]. It might be important to identify patients with early, asymptomatic CAD and potentially early myocardial fibrosis to halt progression to overt HF, even more so as therapeutic options for diastolic dysfunction are still limited to date [7]. Echocardiographic parameters of diastolic dysfunction are commonly not sensitive enough to detect cardiac fibrosis early in asymptomatic patients.

The leucine-rich α-2-glycoprotein-1 (LRG1) is a 50 kDa glycosylated protein consisting of 20–30 amino acid residues that are rich in leucine. Generally, LRG1 expression increases acutely in response to inflammation, thus serving as a biomarker of inflammatory conditions [8]. In addition to inflammation, LRG1 expression has been implicated in the pathogenesis of atherosclerosis [9]. The mechanistic role of LRG1 in cardiac structural remodeling has recently been elucidated: LRG1 regulates and inhibits cardiac fibrocyte activation and consequently cardiac fibrosis by limiting the profibrotic endothelial signaling cascades of TGF-β1 (transforming growth factor-β1) during cardiac remodeling in animal models [5]. There is, however, a paucity of clinical data in humans on the role of LRG1 and cardiac fibrosis.

We hypothesized that LRG1 might be a useful biomarker to identify patients with diastolic dysfunction. Thus, the objective of this study was to determine the association between LRG1 and DD.

## 2. Materials and Methods

### 2.1. Patient Recruitment and Sample Collection

The study was conducted from 1 August 2019 until 1 March 2020 at a tertiary teaching hospital. The study protocol was approved by the institutional review board, Medical Review and Ethics Committee (MREC), Ministry of Health Malaysia (MREC ID NO 20171126-5850), and conformed with the principles of the Helsinki Declaration. The study prospectively enrolled 94 consecutive patients presenting with chronic ischemic symptoms. Written informed consent was obtained from all patients. Patients were included if they fulfilled the following criteria: presentation with symptoms of chronic coronary ischemia (angina symptoms or exertional dyspnea of at least 2 weeks duration); adults (>18 years old), completion of coronary angiography and transthoracic echocardiography. Patients with renal insufficiency (serum creatinine > 150 µmol/L) or recent or current infection or physical trauma (<6 months) were excluded. Patients with symptoms of overt heart failure were excluded. Demographic and clinical characteristics were documented using the hospital’s Electronic Medical Records (EMR) system. Blood samples were collected prior to angiography.

### 2.2. Angiographic and Echocardiographic Assessment

Coronary atherosclerosis was classified into 3 categories: “None”: patent, non-diseased coronary arteries; “non-obstructive”: presence of luminal irregularities with a maximum diameter of stenosis of 70% in at least one major epicardial artery; “Obstructive”: luminal narrowing of more than 70% in at least 1 major epicardial artery. The SYNergy between percutaneous coronary intervention with TAXus and cardiac surgery (SYNTAX) score quantifies the extent of coronary vascular disease and was calculated for each patient by 2 cardiologists blinded to the study’s clinical and angiographic outcomes [1]. Diastolic dysfunction was classified and graded according to the latest societal guidelines of the American Society of Echocardiography and the European Association of Cardiovascular Imaging [2,10,11]. Imaging included apical two- and four-chamber views, from which left atrial (LA) and left ventricular (LV) volumes were measured using the method of discs. LA volume index (LAVi) and LVEF were calculated based on those. Pulsed-wave Doppler of the mitral inflow at the level of valve leaflet tips was used to measure the peak early (E-wave) and late (A-wave) diastolic flow velocities. E/A ratios were calculated. Pulsed-wave Doppler tissue imaging was performed with the sample volume at the lateral and septal mitral annulus to obtain early diastolic annular (e′) velocity. (E/e′) ratios were calculated. Peak velocity of the tricuspid regurgitant jet via continuous-wave Doppler and inferior vena cava diameter and respiratory variation as an estimate of right atrial pressure were acquired. Four variables were analyzed to assess diastolic dysfunction: average E/e’ ratio (>14); septal e’ velocity (<7 cm/s) or lateral e′ velocity (<10 cm/s); peak TR velocity (>2.8 m/s; and LA maximum volume index (>34 mL/m^2^). LV diastolic dysfunction was present if more than half of those variables had values exceeding their respective cut-off values. Patients with reduced LV systolic function (defined as an ejection fraction of less than 50%) or with significant valvular heart disease were excluded from the study.

### 2.3. Other Measurements

Resting blood pressures were recorded on admission. The cut-off for hypertension was 140/90 mmHg. Serum total cholesterol, high-density lipoprotein (HDL) cholesterol, and low-density lipoprotein (LDL) levels were measured using an automated autoanalyzer (Roche Diagnostics, Basel, Switzerland). Patients were categorized as having hyperlipidemia if the LDL cholesterol levels were higher than 3.0 mmol/L or the patients were receiving lipid-lowering medications. Glycated hemoglobin (HbA1c) was measured by a point-of-care immunoassay analyzer (Siemens, Munich, Germany). Patients with HbA1c levels of 6.5% or higher or patients on treatment with antidiabetic drugs were categorized as diabetic. Estimated glomerular filtration rate (eGFR) and urinary creatinine were measured using commercial assays (Immulite, Siemens, Erlangen, Germany). Plasma LRG1 levels were measured using commercially available ELISA kits (Immuno-Biological Laboratories, Fujioka, Japan) according to the manufacturer’s protocol. The measurement range of the LRG1 assays is from 1.56 to about 100 ng/mL with a sensitivity of 0.17 ng/mL.

### 2.4. Statistical Analysis

Data distribution was tested for normality with the Shapiro–Wilk Test, skewness, and kurtosis. Continuous variables are presented as median and interquartile range (if non-normally distributed) and as mean ± standard deviation (if normally distributed). Categorical variables are described as frequencies (percentage). To compare groups with normal distributions, the independent-sample t-test was used, while the Mann–Whitney U test was used for those with non-normal distributions. Categorical data were analyzed with the χ2 test. Prior to multivariate analysis, univariate analyses were employed to evaluate the association between all variables and DD, and potential covariates were selected for adjustment into multivariate analysis. Multicollinearity diagnostics were performed for all covariates using the variance inflation factor (VIF) before including them in multivariate binary logistic regression analyses using the forward selection method. Receiver operating characteristic (ROC) curves assessed the predictability of LRG1 in identifying diastolic dysfunction and the predictive performance of the regression models. Estimated areas under the ROC curve (ROC-AUC) with 95% Cis were used for the evaluation. The Youden Index measures the effectiveness of a diagnostic marker and enables the selection of an optimal threshold value for that marker. The Youden Index was used to determine the optimum cut-off point from the ROC curve, calculated with the formula YI = (sensitivity ± specificity) − 1. All statistical tests were performed two-tailed. A *p*-value less than 0.05 was considered statistically significant. All statistical analyses were performed with SPSS software (version 26.0; SPSS Inc., Chicago, IL, USA) and Medcalc software (version 20.009, Ostend, Belgium).

## 3. Results

### 3.1. Patient Demographics and Clinical Characteristics

A total of 94 consecutive patients (58 males and 36 females) were enrolled in the study. Forty-seven patients fulfilled the echocardiographic criteria for DD, another 47 patients did not. The equal number of patients in the DD and No-DD groups was not intended and occurred by chance. Prior to testing the relationship between LRG1 and DD in this study, a bivariate approach was applied to evaluate the relationship between other demographic and clinical variables. There was no significant difference between the two groups with regard to demographic variables (Table 1). Group comparison of the echocardiographic indices using the Mann–Whitney U test confirmed significant differences (*p* < 0.001) between the No-DD and DD groups. The large effect size for all echocardiographic parameters confirmed the appropriate categorization of patients into the respective groups (Table 2). A Mann–Whitney U test compared plasma LRG1 levels between No-DD (median = 8 ng/mol) and DD groups (median = 14 ng/mol) groups (*p* < 0.001). Plasma LRG1 levels were significantly (U = 417.00, *p* < 0.001) higher in the DD group (Mean rank = 14), 16.0, compared to the No-DD group (Mean rank = 8), 8.6. There was also a higher SYNTAX score in the DD group (Mean score = 24.5), 22.5, compared with the No-DD group (Mean score = 14.8), 10.4, *p* < 0.001 (Appendix A).

### 3.2. Regression Model Analysis of LRG1 for Diastolic Dysfunction

Binary logistic regression was employed to evaluate the relationship between LRG1 and DD. Based on bivariate analysis, several variables including number of coronary lesions, SYNTAX, urea, creatinine, diabetes mellitus, and low-density lipoprotein (LDL) levels showed a significant association with DD. Therefore, the final model was adjusted considering all these variables using stepwise regression (forward-conditional method). In the final model, only three factors comprising SYNTAX, creatinine, and diabetes mellitus (DM) remained as significant variables. The results of both unadjusted and adjusted model are presented in Table 3. LRG1 had significant predictability of DD with an OR = 1.26 (95% CI: 1.23–1.42), indicating that for every one-unit increase in LRG1, a 26% increase in the odds of DD being present is to be expected. After adjustment, the odds ratio increased to OR = 1.32 (95% CI: 1.14–1.53). Furthermore, the accuracy of LRG1 for the prediction of DD was evaluated with ROC-AUC. Figure 1 showed the ROC curves for LRG1 with an AUC = 0.89 (95% CI: 0.82–0.95) of the model (Z = 2.356, *p* < 0.001) after adjustment with SYNTAX, creatinine, and diabetes mellitus (DM). The ROC for the DD model was calculated and resulted in a significantly improved accuracy (Table 4). The highest Youden Index (YI) was calculated as YI = 0.503 (95% CI: 0.351–0.643). The cut-off value of “9” LRG1 obtained the highest Youden Index, with 78% sensitivity (95% CI: 65.3–87.7) and 72.3% specificity (95% CI: 57.4–84.4) for predicting DD.

## 4. Discussion

This study investigated the association between LRG1 and left ventricular diastolic dysfunction (LVDD). We demonstrated that patients with LVDD had significantly higher circulating LRG1 levels compared to patients without LVDD. We found that LRG1 was independently and highly predictive for LVDD, although the cross-sectional research design makes it impossible to determine the causal effect.

LVDD increases the risk of adverse cardiovascular events and commonly progresses to symptomatic heart failure with preserved ejection fraction (HFpEF) [12,13,14,15]. The prevalence of HFpEF is increasing due to an aging population, increased awareness, refined diagnostic criteria, and advances in imaging [16,17]. In fact, symptomatic patients with echocardiographic evidence of diastolic dysfunction, previously subsumed as having “diastolic heart failure”, are now being classified as having HFpEF. This is a major public health issue as it accounts for more than half of the total heart failure prevalence [16]. Despite its clinical importance, there is no reference strategy for the diagnosis of HFpEF; even more challenging is the identification of patients with early stages of LVDD who might display very few symptoms [18]. During the early stages of HFpEF, echocardiographic parameters can be normal at rest and the diastolic dysfunction is only revealed during exercise stress testing. Likewise, natriuretic peptides are frequently within the normal range or only minimally raised [19,20]. Diagnosis at early stages of diastolic dysfunction could identify patients that would benefit from early intervention with lifestyle modifications, intensification of treatment regimes, and closer follow-up. Hence, there is an urgent need for a more sensitive marker of diastolic dysfunction.

HFpEF is not caused by a single pathological process but is rather a complex disease, with multiple underlying pathophysiological mechanisms. It has been postulated that the plethora of comorbidities that are commonly present in HFpEF patients creates a proinflammatory state resulting in generalized endothelial inflammation. Reduced nitric oxide bioavailability with negative effects on cyclic GMP content and protein kinase G activity in adjacent myocytes results in fibrosis, diastolic LV stiffness, and HF development [21]. Micro- and macrovascular ischemia has been shown to be a major determinant of diastolic dysfunction [22,23,24,25,26]. Inflammation plays a key role in the pathogenesis of atherosclerosis and promotes deposition of LDL in the affected vasculature and formation of atherosclerotic plaques.

The role of LRG1 as a key regulator of vascular disease associated with inflammation is emerging, as studies have reported an increased expression of LRG1 in such a setting [27,28,29,30,31]. LRG1 has been shown to be elevated in animal models of retinopathy [32], in patients with coronary artery disease [30], in arterial stiffness [33], and, most recently, in animal models of cardiac fibrosis [34]. It has been proposed that LRG1 is upregulated in cardiac myocytes during fibrotic cardiac remodeling [32,34,35]. Liu et al. demonstrated recently that overexpression of LRG1 attenuated cardiac fibrosis and overt heart failure in the myocardium of heart failure animal models [34]. As such, elevated levels of LRG1 likely reflect the physiological compensatory response to a fibrotic process. LRG1 exerts its regulatory anti-fibrotic effect by inhibiting the inflammatory signaling of the transforming growth factor (TGF)-β1 within a complex molecular network involving PPAR (peroxisome proliferator-activated receptor) β/δ and SMRT (silencing mediator for retinoid and thyroid hormone receptor) [36,37].

The SYNTAX score is a scoring algorithm that quantifies the extent of coronary atherosclerosis. It is widely used to risk stratify patients in clinical practice and to support decision-making [38]. We utilized the SYNTAX score as a surrogate marker of the atherosclerotic burden in our study cohort. Pairwise comparison demonstrated, in addition to the elevated LRG1 levels, markedly higher SYNTAX scores in the DD group. The association of LRG1 and new-onset atherosclerotic cardiovascular disease has recently been reported in a cohort of the Framingham Heart Study [39]. It has been proposed that the elevation of LRG1 occurs at early, preclinical stages [29,39].

The potential role of LRG1 in heart failure was first reported by Watson et al. [35], who studied a heterogeneous group of patients with overt systolic and diastolic heart failure. He found that LRG1 was consistently overexpressed in high BNP serum and identified heart failure patients independent of BNP. Our study confirmed Watson’s experimental findings in a clinical setting by demonstrating a potentially mechanistic relationship between LRG1, diastolic dysfunction, and the extent of vascular disease.

The multivariable logistic regression model showed that LRG1 had significant predictability of DD. The overall high diagnostic accuracy of the regression models in our study was confirmed with ROC analysis. The ROC for the adjusted model demonstrated a high accuracy with an AUC = 0.89 (95% CI: 0.82–0.95). As seen in Figure 1, nominally, the OR for predicting DD was higher for diabetes than for LRG1. Diabetes is, however, a risk factor for the pathogenesis of DD and is prevalent in about one-fifth of adults in the population from which the study sample was derived. Diabetes would not be a good diagnostic marker of the disease itself in a setting of high prevalence. In contrast, LRG1 could become a valid biomarker for DD if our findings can be reproduced in well-designed further studies.

The limitation of our study is the relatively small sample size and the cross-sectional study design. Inclusion of patients with ischemic symptoms at hospital presentation may create a selection bias potentially limiting the applicability of the findings to a subset of patients with symptomatic ischemia. The current study did not test for other biomarkers such as brain natriuretic peptides (BNPs) and High Sensitivity C-Reactive Protein (hsCRP) [40]. Overall, we consider this research preliminary and encourage replication.

In summary, the progression from subtle impairment of diastolic function to clinically overt heart failure is a slow and insidious process. Hence, the discovery of LRG1 as a sensitive and independent marker of diastolic dysfunction might open new avenues for earlier and more reliable diagnoses of DD. Further prospective studies with larger cohorts of patients at varying stages of asymptomatic DD and symptomatic heart failure, as well as healthy controls, are warranted to validate the clinical utility of LRG1 as a novel biomarker for risk stratification and prognostic evaluation.

## 5. Conclusions

Plasma LRG1 is suggested to be a sensitive and independent indicator of diastolic dysfunction.

## Figures and Tables

**Figure 1 biomedicines-11-00944-f001:**
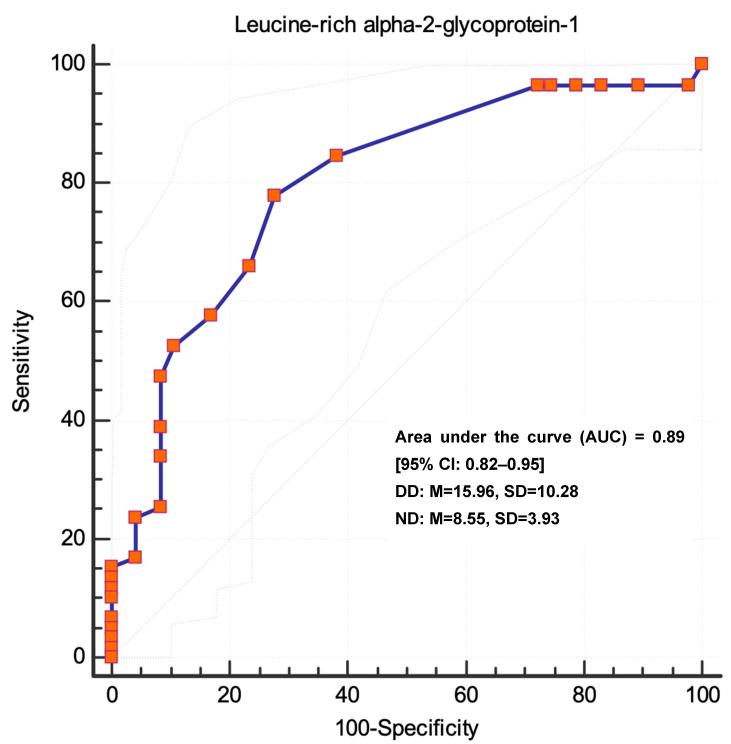
Receiver Operator Characteristic (ROC) curve of LRG1 in patients with DD. Diagnostic accuracy for DD is presented as Mean Area Under Curve (AUC) = 0.89 [95% CI: 0.82–0.95], *p* < 0.001), after adjustment with SYNTAX, creatinine, and diabetes mellitus (DM). Butted line indicates reference line.

**Table 1 biomedicines-11-00944-t001:** Baseline Demographics and clinical data between patients without diastolic dysfunction (No-DD) and with diastolic dysfunction (DD).

Variable	Diastolic Function		*p*-Value
	No-DD (n = 47)	DD (n = 47)	
Age (years)	62.04 ± 11.35	63.53 ± 12.16	0.323
Gender			
Male	27 (57.40)	31 (66.00)	0.396
Female	20 (42.60)	16 (34.00)	
Race			
Malay	14 (29.80)	18 (38.3)	0.228
Chinese	21 (44.7)	13 (27.7)	
Indian	12 (25.50)	16 (34.0)	
Number of coronary lesion(s)			
0	8 (17.00)	4 (8.50)	<0.001 **
1	20 (42.60)	3 (6.40)	
2	11 (23.40)	10 (21.30)	
3	8 (17.00)	30 (63.80)	
Diastolic dysfunction grading			
Grade 1	NA	38 (80.9)	
Grade 2	NA	5 (10.6)	
Grade 3	NA	4 (8.5)	
LRG1 levels (ng/mL)	8 (4)	14 (8)	<0.001 **
SYNTAX	7 (17)	24.5 (15)	<0.001 **
Laboratory tests			
Hemoglobin (g/dL)	13.2 (1.5)	13.3 (1.5)	0.758
Urea (mmol/L)	5.5 (3.6)	6.6 (3.9)	0.037 *
Creatinine (mcmol/L)	78 (28)	90 (48)	0.018 *
Risk Factors			
Diabetes Mellitus			
Non-diabetic	38 (80.9)	26 (55.3)	0.008 **
Diabetic	9 (19.1)	21 (44.7)	
Lipid Profile			
Total cholesterol (mmol/L)	4 (1.8)	4.5 (1.3)	0.117
Triglyceride (mmol/L)	1 (0.8)	1.2 (0.7)	0.604
LDL (mmol/L)	2.24 (1.33)	3 (1.47)	0.031 *
HDL (mmol/L)	1.07 (0.5)	1.08 (0.38)	0.678
Hyperlipidemia			
No	40 (85.10)	37 (78.70)	0.421
Yes	7 (14.90)	10 (21.30)	
Hypertension			
<140/90 mmHg	27 (57.4)	31 (66.0)	0.396
>140/90 mmHg	20 (42.6)	16 (34.0)	
Smoking			
Non-smoker	24 (51.10)	27 (57.40)	0.204
Current smoker	20 (42.60)	20 (42.60)	
Ex-smoker	3 (6.40)	0 (0.00)	
Medication use			
Perindopril	27 (57.4)	21 (44.7)	0.216
Beta Blocker	26 (55.3)	20 (42.6)	0.302
Frusemide	6 (12.8)	2 (4.3)	0.139
Spironolactone	0 (0)	1 (2.1)	0.315
Metformin	9 (19.1)	19 (40.4)	0.024 *
Gliclazide	3 (6.4)	7 (14.9)	0.181
Insulin	6 (12.8)	8 (17.0)	0.562

Categorical variables are frequency, n (%). Continuous variables are median (IQR). Pairwise groups comparison was performed by Pearson’s chi-square test for categorical variables, or Mann–Whitney U test for continuous variables, * *p* < 0.05, ** *p*< 0.005. No-DD = without diastolic dysfunction, DD = with diastolic dysfunction.

**Table 2 biomedicines-11-00944-t002:** Comparison of echocardiographic indices between patients without diastolic dysfunction (No-DD) and with diastolic dysfunction (DD).

Index	No-DD	DD	*p* Value	Effect Size
	Median (IQR)	Median (IQR)		
LAVI	28 (3)	35 (2)	<0.001	1.54
TR velocity	2.1 (0.4)	2.9 (0.1)	<0.001	2.7
Septal e′	7.8 (1.5)	6.7 (1.5)	<0.001	1.42
Lateral e′	12 (3.3)	9.8 (1.1)	<0.001	1.33
E/e′	12 (2)	15 (1)	<0.001	1.64
EF (%)	67 (21)	60 (14)	0.069	0.8

LAVI—left atrial volume index, TR velocity—tricuspid valve regurgitant jet velocity, EF—ejection fraction, DD—diastolic dysfunction, IQR—interquartile range, E—early mitral inflow velocity, e′—early diastolic mitral annular tissue velocity.

**Table 3 biomedicines-11-00944-t003:** Univariable and multivariable logistic regression analyses of plasma LRG1 for prediction of DD.

Variables	Unadjusted OR	*p*-Value	Adjusted OR	*p*-Value
	[95% CI]		[95% CI]	
LRG1	1.26	<0.001	1.32 **	<0.001
	[1.23–1.42]		[1.14–1.53]	
SYNTAX	1.1	-	1.08 **	0.007
	[1.06–1.16]		[1.02–1.14]	
Creatinine	1.02	-	1.04 **	0.005
	[1.01–1.04]		[1.01–1.06]	
DM	3.41	-	6.93 **	0.006
	[1.35–8.61]		[1.76–27.34]	

The odds ratio (OR) compared the probability for a positive DD diagnosis and is presented as unadjusted and adjusted for SYNTAX, creatinine, and DM, with 95% CI indicating clinical significance. *p*-values of ** *p* < 0.001 were statistically significant. The adjusted model (χ2(4) = 57.866, *p* < 0.001) explained 57.8% of the variance in DD and correctly classified 83% of cases. DD, diastolic dysfunction; OR, odd ratio; 95% CI, 95% confidence interval.

**Table 4 biomedicines-11-00944-t004:** Comparison of Predictive performance of regression models.

Predictive Performance of Regression Model				Pairwise Comparison of ROC Curves	
Model	AUC [95% CI]	Standard Error	*p*-Value	z	*p*-Value
Unadjusted Model	0.79 [0.70–0.87]	0.04	<0.001 ***	2.356	0.0185
Adjusted Model	0.89 [0.82–0.95]	0.03	<0.001 ***		

*** *p* < 0.001.

## Data Availability

The data generated during and/or analyzed during the current study are available from the corresponding author upon reasonable request.

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
