# Peer review of "Leucine-Rich Alpha-2-Glycoprotein: A Novel Predictor of Diastolic Dysfunction"

_biomedicines, 2023, doi:10.3390/biomedicines11030944_

Round 1

Reviewer 1 Report

see attached file.

Author Response

Please find our response to the reviewers and the modified manuscript, under attachment for "author to respond reviewer_(1) - MDPI.docx".

We are sorry to have taken some time, but the team works in 3 countries and we needed some time to retrieve technical details requested by the reviewers. Thank you for your understanding. 

Reviewer 2 Report

The authors evaluated the association between leucine-rich α2-glycoprotein (LRG1) as a mediator of cardiac fibrocyte activation inducing myocardial fibrosis, and the left ventricular diastolic dysfunction (LVDD) in a cross-sectional study. They found that patients with LVDD had significantly higher values of circulating LRG1 compared with patients without LVDD. They also found that LRG1 can predict LVDD with a good sensibility and specificity. This information is interesting and useful and might have important implications for diagnosis and possibly even for future treatment of LVDD. The article is well written, the introduction offers a proper background information, the methods and results are adequately and clearly described and presented. However, I have some comments/suggestions:

    - the authors hypothesise that LRG1 might be useful to identify patients with asymptomatic LVDD, highlighting the importance of finding a sensitive marker to detect/diagnose the early stages, even before echocardiographic evidence. Although there is a high probability for LRG1 to be such a marker, this study demonstrates its good prediction for symptomatic LVDD with echocardiographic evidence. Thus, in order to avoid misleading information, I would suggest for the authors to rephrase the hypothesis and the corresponding discussion paragraph.

    - there is no information about patients with significant valvular disease. If they were included in the study the authors should describe how they established the LVDD echocardiographic diagnosis (for example in significant mitral regurgitation/stenosis). If they were excluded from the study, the authors should mention significant valvular disease among the exclusion criteria.  

 - please carefully check for typos.

 Overall, I recommend this paper to be published after minor revisions.

Author Response

Please find our response to the reviewers and the modified manuscript, under "author to respond reviewer_(2) - MDPI". We are sorry to have taken some time, but the team works in 3 countries and we needed some time to retrieve technical details requested by the reviewers. Thank you for your understanding. 

Reviewer 3 Report

I am grateful for the opportunity to review the interesting manuscript of Loch et al. "Leucine Rich Alpha-2-Glycoprotein: A Novel Predictor of Diastolic Dysfunction". In this article, the authors investigated the association of LRG1 and left ventricular diastolic dysfunction (LVDD). They demonstrated that patients with LVDD had significantly higher circulating LRG1 levels compared to patients without LVDD. They also found that LRG1 was independently and highly predictive for LVDD in binary logistic regression analysis. Consequently, it becomes possible to use LRG1 as a sensitive and independent marker of diastolic dysfunction.

However, when reviewing the manuscript, I had questions and comments to which I would like to receive answers from the authors of the article.

1. It is noteworthy that the number of patients in the groups is the same (47 patients each). It is clear that with the sequential inclusion of patients in the study, such an equal distribution of patients into groups with the presence / absence of diastolic dysfunction of the left ventricle cannot be. Accordingly, the authors carried out some kind of selection of patients, but about this in section 2.1. Patient recruitment and sample collection nothing listed. Therefore, the presence of bias due to this nature of the inclusion of patients cannot be ruled out. It is necessary to describe in more detail the nature of the inclusion and selection of patients in the study (optimally, provide a flowchart). The fact of such selection of patients should also be reflected in the Study Limitations section.

2. In section 2.2. Angiographic and echocardiographic assessment authors should indicate the criteria for left ventricular diastolic dysfunction that they used in their article.

3. In the first paragraph of the Discussion section, the authors indicated that "LRG1 could be a suitable marker for the early identification of diastolic dysfunction in at-risk patients as well as for the monitoring of disease progression and treatment effects". However, the results of the authors do not confirm this assumption in any way, and one should speak less categorically about such a possible use of this marker.

4. In the Discussion section, the authors note that "Nominally the OR for predicting DD was higher for diabetes than for LRG1. Diabetes is however a risk factor for the pathogenesis of DD and prevalent in about one fifth of adults of the population from which the study sample was derived. Diabetes would not be a good diagnostic marker of the disease itself in a setting of high prevalence. In contrast, LRG1 could become a valid biomarker for DD...". However, the LRG1 biomarker itself is not specific for the diagnosis of diastolic left ventricular dysfunction and even myocardial fibrosis. As shown in recent reviews (1,2), LRG1 is elevated in a wide range of diseases, LRG1 is a pleiotropic and pathogenic signaling molecule. Based on this, in this article, it is necessary to conduct an additional analysis on the association of the LRG1 level with various clinical indicators (diabetes, severity of coronary artery disease, creatinine level), and not only with the presence of left ventricular diastolic dysfunction. And the second. Since an increase in the LRG1 level can occur, for example, in cancer, lung diseases, CNS pathology, in each specific case we cannot associate this increase with the presence of left ventricular diastolic dysfunction without its echocardiographic signs. Then a natural question arises - is there a need for an additional definition of a new biomarker?

References:

1.     Camilli C, Hoeh AE, De Rossi G, Moss SE, Greenwood J. LRG1: an emerging player in disease pathogenesis. J Biomed Sci. 2022 Jan 21;29(1):6. doi: 10.1186/s12929-022-00790-6.

2.     Zou Y, Xu Y, Chen X, Wu Y, Fu L, Lv Y. Research Progress on Leucine-Rich Alpha-2 Glycoprotein 1: A Review. Front Pharmacol. 2022 Jan 5;12:809225. doi: 10.3389/fphar.2021.809225.

Author Response

Please find our response to the reviewers and the modified manuscript, under the "attachment for "author to respond reviewer_(3) - MDPI.docx". We are sorry to have taken some time, but the team works in 3 countries and we needed some time to retrieve technical details requested by the reviewers. Thank you for your understanding.

Round 2

Reviewer 3 Report

The authors answered questions and made some changes to the text of the manuscript. However, I was not satisfied with the answer of the authors to my question 2. Firstly, at the time of the second review, I do not see on the site either the question of another reviewer, or the answer of the authors to it. Secondly, the essence of the answer. The authors still retained the phrase in the beginning of Discussion "LRG1 could be a suitable marker for the identification of diastolic dysfunction in at-risk patients as well as for the monitoring of disease progression and treatment effects". You can agree with the first statement, but not with the second. The authors did not study disease progression or treatment effects. Therefore, these assumptions should be accompanied by reservations that such use of this biomarker deserves attention, but requires confirmation in further studies. And these arguments should not be placed at the beginning of the "Discussion" section, but closer to the end, where the authors discuss the possible clinical applications of this method.

Author Response

Thank you for your comments. We apologize for overlooking this and indeed one of the other reviewers had made a similar comment. 

We changed the statement to: ‘We found that LRG1 was independently and highly predictive for LVDD, although the cross-sectional research design makes it impossible to determine the causal effect.’ The statement  "LRG1 could be a suitable marker for the identification of diastolic dysfunction in at-risk patients as well as for the monitoring of disease progression and treatment effects" was removed from the first paragraph.

We have left the final paragraph as is: “… the discovery of LRG1 as a sensitive and independent marker of diastolic dysfunction might open new avenues for earlier and more reliable diagnoses of DD. Further prospective studies with larger cohorts of patient at varying stages of asymptomatic DD and symptomatic heart failure, as well as healthy controls, are warranted to validate the clinical utility of LRG1 as a novel biomarker for risk stratification and prognostic evaluation.”

We hope this summarizes our novel findings while accepting that this a very preliminary study which needs to be replicated and expanded through future research.

Enclosed is the revised manuscript, marked up using the track changes.

Thank you.